# PeerJ

# When is an ecological network complex? Connectance drives degree distribution and emerging network properties

Timothée Poisot and Dominique Gravel

Université du Québec à Rimouski, Département de Biologie, Rimouski (QC), Canada
Québec Centre for Biodiversity Sciences, Montréal (QC), Canada

## ABSTRACT

Connectance and degree distributions are important components of the structure of ecological networks. In this contribution, we use a statistical argument and simple network generating models to show that properties of the degree distribution are driven by network connectance. We discuss the consequences of this finding for (1) the generation of random networks in null-model analyses, and (2) the interpretation of network structure and ecosystem properties in relationship with degree distribution.

## INTRODUCTION

Ecologists developed a strong interest for network theory as it allows them to make sense of some of the complexity of ecological communities. In contrast to early approaches on "community modules" (groups of a few species within a large community, *Holt, 1997*) a network level approach allows one to account for the whole community scale (*Dunne, 2006*), thus integrating all direct and indirect interactions (*Berlow et al., 2009*). Ecological networks have often been called "complex" (*Williams & Martinez, 2000*), on account of the fact that they represent objects (ecological communities) with complex dynamics (i.e., non-linear, sensitive to indirect interactions). Because networks are multi-faceted objects with a rich range of structure, ecologists have been looking for emerging properties that can be easily measured and analyzed, and that relate to ecological properties and processes.

Early in the ecological network literature, connectance, i.e., the proportion of realized ecological interactions among the potential ones (most often the squared species richness), has been recognized as a central network property (*May, 1972*; *Yodzis, 1980*; *Martinez, 1992*). In part, this success can be attributed to the relationship between connectance and early definitions of network complexity (*Pimm, 1982*), and to the fact that connectance predicts reasonably well key dynamical properties of ecological networks (*Dunne, Williams, & Martinez, 2002a*, *Dunne, Williams, & Martinez, 2002b*) including their stability (*May, 1972*). More recently, attention shifted from connectance, a community-averaged property, to the degree distribution, that is the statistical properties

Corresponding author
Timothée Poisot, t.poisot@gmail.com

of the distribution of number of interactions per species. Variation of degree distribution among networks has often been taken as evidence that assembly or interaction mechanisms differ (*Vázquez, 2005*; *Williams, 2011*), and increasingly refined methods to estimate degree distribution have been devised (*Williams, 2009*). Some authors proposed that degree distribution, rather than connectance, is driving higher level network properties such as nestedness or modularity, which are important drivers of network dynamics (*Fortuna et al., 2010*).

However, it is worth asking if we were not too quick in focusing most of our research effort on degree distribution, in detriment to more fundamental work on connectance and its effects. A network, ecological or otherwise, can be viewed as a physical space that edges (interactions) occupy. The size of this space is limited by the number of nodes. This means that there are physical constraints on the filling of a network, due to the fact that placing the first edge will limit the number of ways to place the remaining edges, and so on. For example, there is only one way to have a fully connected network, and there are a limited number of ways to have a network with the lowest possible connectance. For this reason, and given the rising importance of degree distribution in the literature, it is important that we clearly understand how constrained this distribution actually is in relation to connectance. In this contribution, using an argument from combinatorial statistics and simulations of pseudo-random networks under two different models, we present strong evidence that degree distribution, along with other emerging network properties, are constrained (and can be predicted to a certain extent) by connectance. We discuss the consequences of our results for the comparison of different ecological networks, and for the generation of random networks in null-model analyses.

## STATISTICAL ARGUMENT

Assuming an ecological network made of $n$ species, and assuming undirected interactions with no self-edges (e.g., no species can interact with itself), there can be at most $M = n(n-1)/2$ interactions in this network, in which case it is a complete graph (the results presented below hold qualitatively for both directed graphs, and graphs in which self-edges are allowed). We note this maximal number of links $M_n$. With this information in hand, it is possible to know the total number of possible networks given a number $l$ of interactions. Ecologists are often more familiar with networks being represented as their adjacency matrix, i.e., (with minimal simplifications) a matrix with as many rows and columns as there are species, and a 1 at the intersection of two species that interact. In an undirected network, the existence of an edge between species $A$ and $B$ imply *two* interactions (i.e., $A \rightarrow B$ and $B \leftarrow A$), and so assuming no self-edges, the total number of ones in the adjacency matrix of a complete, undirected graph, is $n(n-1)$. Throughout this paper, we represent networks as graphs, and not as adjacency matrices. We consider only the situation of unipartite networks, i.e., that can be represented by a square matrix. While the shape of the matrix (i.e., the ratio of columns numbers over rows numbers) will have an impact on the results, we conducted preliminary simulations showing that the results hold qualitatively in bipartite networks of varying shapes.

If we term $S_n$ the set of all possible $M_n$ edges in a $n$-node network, then the number $G_{n,l}$ of possible networks with $l$ links is the number of $l$-combinations of $S_n$, i.e., how many possibilities are there to pick $l$ edges among $M_n$. Formally, this is expressed as $G_{n,l} = C_l^{M_n}$, (where $C_x^y$ is the binomial coefficient, i.e., the number of possible ways to pick $x$ elements among a set of $y$ elements) or

$$G_{n,l} = \frac{M_n!}{l!(M_n - l)!}$$

Note that this number of possible networks include some graphs in which nodes have a degree of 0, and that in most ecological studies, such nodes will be discarded. We therefore have to evaluate how many of such networks will be found within $G_{n,l}$. In addition, in a null-model context (*Bascompte et al., 2003*; *Fortuna & Bascompte, 2006*), having unconnected nodes in random replicates will change the richness of the community, thus possibly biasing the value of randomized emerging properties. As some measures of network structure covary with species richness, if one is to generate a randomly expected distribution of the values of these properties, then it is important to hold species richness constant. Finding out the number of networks in which a given node has a degree of 0 is similar to finding out how many networks exist with $l$ links between the $n - 1$ nodes. If one node is removed from the network, there are $C_{n-1}^n$ possible combinations of nodes (which is $(n)!/((n-1)!(n-(n-1))!)$, which further simplifies to $n$). For each of these, there are $G_{n-1,l}$ possible networks configurations. Note that these networks will also include situations in which *more* than one species has a degree of 0, so that by recurrence, evaluating $G_{n-2,l}$ and so forth is not necessary (all networks with more than one node of null degree are within the set of the networks with at least one node of null degree). Calling $R_{n,l}$ the number of networks with $n$ nodes and $l$ edges in which all nodes have at least one edge attached, we can write that the number of networks with all nodes having at least one edge is the total number of networks minus the number of networks having at least one node of null degree (evaluated for each node), or

$$R_{n,l} = G_{n,l} - C_{n-1}^n \times G_{n-1,l}$$

We call the quantities $R$ and $G$, respectively, the *realized* and *total* network spaces. They measure how many networks of $n$ nodes and $l$ edges exists, either allowing or preventing the existence of nodes with no interactions. Based on this reasoning, we can make two predictions.

### Prediction 1:
Because $C_x^y = C_{y-x}^y$, it comes that the total network space is largest when $l = M_n/2$. As in this context the maximal number of edges is $M_n$, we define effective connectance as $Co = l/M_n$, so $\max(G_{n,l})$ is reached at $Co = 1/2$. The algebraic expression of the maximum value of $R_{n,l}$ is hard to find, but simulations show that it also occurs around $Co = 1/2$. In other words, regardless of the number of nodes in a network, the "degrees of freedom" of network structure, as indicated by the size of the realized and total network spaces, is maximized at intermediate connectance.

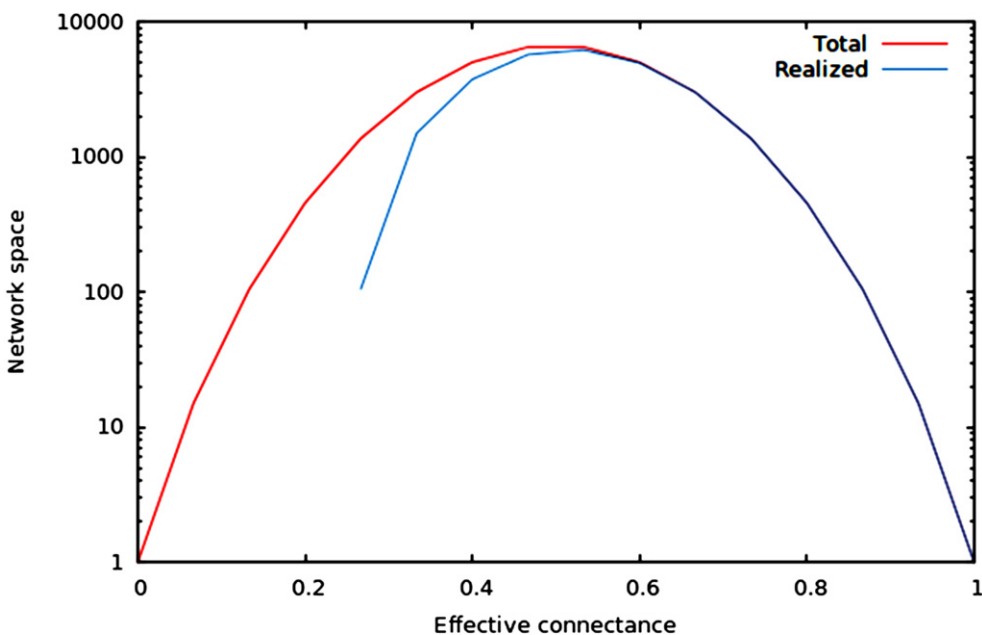

**Figure 1 Size of the total and realized network space for $n = 6$.** As predicted in the main text, (1) the size of network spaces peaks at $Co = 1/2$, and (2) the size of the realized network space becomes asymptotically closer to the size of the total network space when connectance increases.

## Prediction 2:

$R_{n,l}$ will become asymptotically closer to $G_{n,l}$ when $l$ is close to $M_n$. In other words, there is only one way to fill a network of $n$ nodes with $M_n$ interactions, and in this situation there is no possibility to have nodes with a degree of 0. In the situation in which $l = M_n$, $G_{n,l} = C_{M_n}^{M_n} = 1$, given that $M_n > M_{n-1}$, it comes that $G_{n,l} = R_{n,l} = 1$. Intuitively enough, this implies that ecological systems in which connectance is high will display very little variation from one another, as far as the distribution of emergent network properties (e.g., variance of the degree distribution, nestedness, . . . ) is concerned.

We now illustrate these predictions using networks of 10 nodes, with a number of edges varying from 10 to $M_{10}$ (i.e., 45 edges). As illustrated in Fig. 1, the size of the network space has a hump-shaped relationship with connectance, and the size of the realized network space becomes closer to the size of the total network space when connectance increases.

In Fig. 2, we show that regardless of the network size, the relative size of the realized network space increases with connectance. The rate at which it occurs increases with network size. However, in all cases, when connectance is low, there are only a very small proportion of the total network space in which all nodes have at least one edge. This suggests that the structure of extremely sparse networks is also strongly constrained. This is congruent with historical findings by *Erdos & Rényi (1959)*, namely that the probability of each node being connected to the graph's largest connected component (i.e., any set of

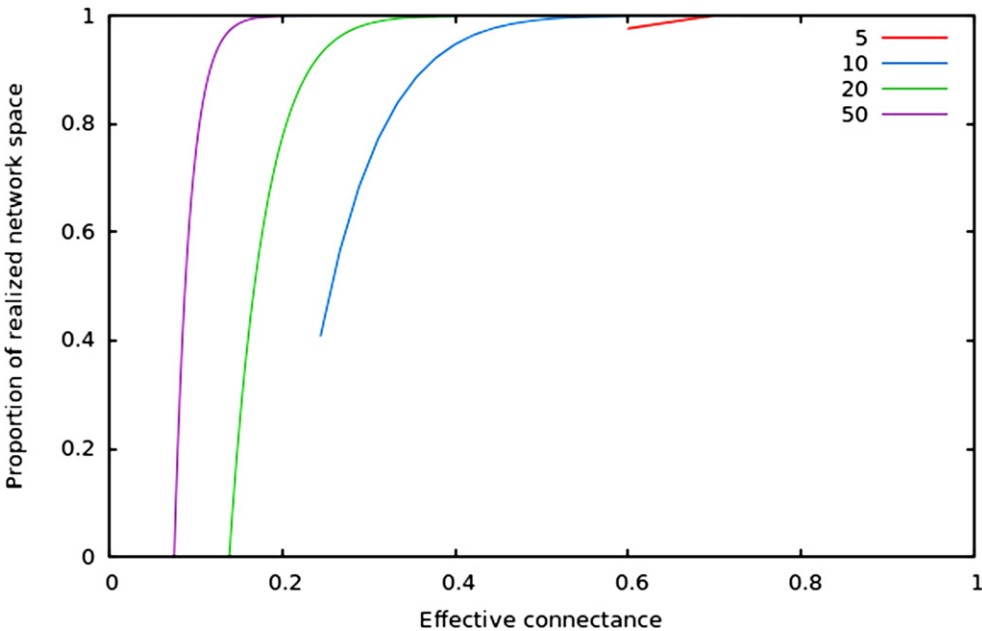

**Figure 2** **Relative size of the realized network space compared to the total network space when connectance increases, for four different network sizes.**

vertices of which any two are connected by at least one path) increases with average degree (thus for high connectances, all nodes are likely to be connected to the giant component, hence no node has a degree of 0). In the context of ecology, in which most networks have a low connectance, it implies that generating random networks to test null hypotheses can be a computationally intensive task, as the realized network space is (proportionally) small.

## SIMULATIONS

In the previous part, we show mathematically that connectance (the number of realized *vs.* possible interactions), relative to the network size, determines the size of the *network space*, i.e., how many possible network combinations exist. Based on this, we can therefore expect that the degree distribution will be contingent upon network connectance. Specifically, we expect that the variance of the degree distribution, which is often related to ecosystem properties and other network structures (*Fortuna et al., 2010*), will display a hump-shaped relationship with connectance. The mean, kurtosis, and skewness of the degree distribution should all vary in a monotonous way with connectance.

In the simulations below, we use networks of 30 nodes, filled with 35 to $M_{30}$ interactions, by steps of 10. We use two different routines to generate random networks that are contrasted in the way they distribute edges among nodes. First, we generate Erdős-Rényi (ER, undirected) graphs, meaning that every potential interaction has the same probability of being realized (*Erdos & Rényi, 1959*). We use an algorithm inspired by

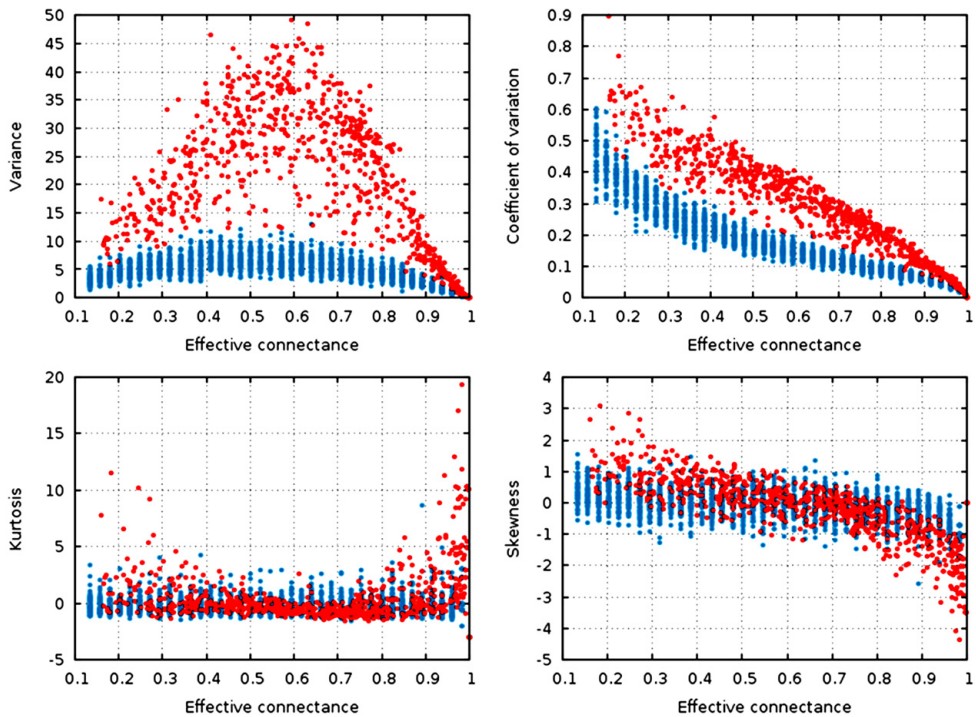

**Figure 3  Statistical descriptors of the degree distribution of randomized networks, $n = 30$, increasing connectance.** These results show that central properties of the degree distribution are contingent upon connectance, at a given network size, and under a given network generation model. ER networks are in blue, niche-model networks are in red. Each point represent a single generated network.

*Knuth (1997)*, allowing to fix the number of edges in the graph rather than the probability of an edge occurring, although the generated graphs have the same properties as the original ER model. A total of 19,000 networks are generated this way. Second, we use the niche model of food webs (*Williams & Martinez, 2000*), which generates (directed) networks under rules representing hypothesized mechanisms of prey-selection in empirical ecosystems (*Gravel et al., 2013*). This particular model assumes that the existence of interactions is constrained by the position of species along a "niche" axis, for example, body size. Other randomization methods for food webs exists, but given that *Stouffer et al. (2005)* showed that they yield similar degree distributions to the niche model, we will not use them here. A total of 500 replicates for each value of $l$ are generated. All networks generated with the two models are free of self-edges and nodes with a null degree.

For each replicate, we measure the degree distribution and report its variance, coefficient of variation, kurtosis, and skewness. In addition, for each network, we fit a power-law distribution on the sorted degree distribution using the least-squares method; we report the power-law exponent.

Qualitatively, the random graphs and the niche networks behave exactly the same. With the exception of the kurtosis, *all* statistical descriptors of the degree distribution

were influenced by the effective connectance (Fig. 3). As predicted in the previous part, variance of the degree distribution is hump-shaped with regard to connectance, which implies that as average degree increases with connectance, the coefficient of variation of the degree distribution decreases at high connectances. Note also that the range of variances in the degree distribution is higher at intermediate connectances, but lower at the extreme. Due to the fact that the Erdős-Rényi graphs we simulate are essentially Poisson random graphs, it is expected that the variance of their degree distribution would be lower than for the niche model, which in contrast *forces* strong difference in the degree of species according to their niche position.

To quantify the impact of connectance on the different network properties, we measured the proportion of variance explained by the linear regression of a given property against connectance (in such cases as had a linear relationship between the two, i.e., all measures but variance). Kurtosis is independent of connectance ($R^2_{niche} = 0.04$, $R^2_{ER} = 0.06$), while skewness decreases with connectance, although more markedly so in the niche model ($R^2_{niche} = 0.66$, $R^2_{ER} = 0.26$). This result is expected. Positively skewed distribution have longer or fatter right tails, indicating mostly low values (low degree): unconnected networks are made mostly of species with a weak generality (*Schoener, 1989*). On the other hand, negative skewness indicate that most of the values in the distribution are high. Ecologically, it means that most species are wide-range generalists, which happens in densely connected networks. This bears important ecological consequences, as it indicates that due to physical constraints acting on the filling of interactions within the graphs, networks with intermediate connectances are expected to have species with both low and high generality (*Schoener, 1989*). The coefficient of variation of the degree distribution is extremely well predicted by connectance alone ($R^2_{niche} = 0.91$, $R^2_{ER} = 0.87$).

The estimate of the power-law exponent increases when connectance increases (Fig. 4, $R^2_{niche} = 0.91$, $R^2_{ER} = 0.70$). This indicates that the degree distribution flattens when connectance increases. Taken with the elements presented above, we show that all of the estimators of the degree distribution vary strongly with connectance of the network. Although power-laws should be truncated as soon as the probability of a species having a degree of $2 \times n$ (or $n$ in undirected networks) is not negligible, and as such the fitting of power-laws should not be done on highly connected networks for practical purposes, this result emphasizes the key role of connectance in driving central network structure properties.

## PRACTICAL CONSEQUENCES

Randomized null models are often used to estimate how much a given network property deviates from its random expectation (*Flores et al., 2011*). Our results show two things. First, except for extremely high or low connectance, the proportion of the network space that will be explored using $10^3$ or $10^4$ replicates (typical values in null models analyses) is orders of magnitude smaller than the *realized* network space. Although this is somewhat compensated by the fact that a part of these networks are isomorphic, the risk of inferring

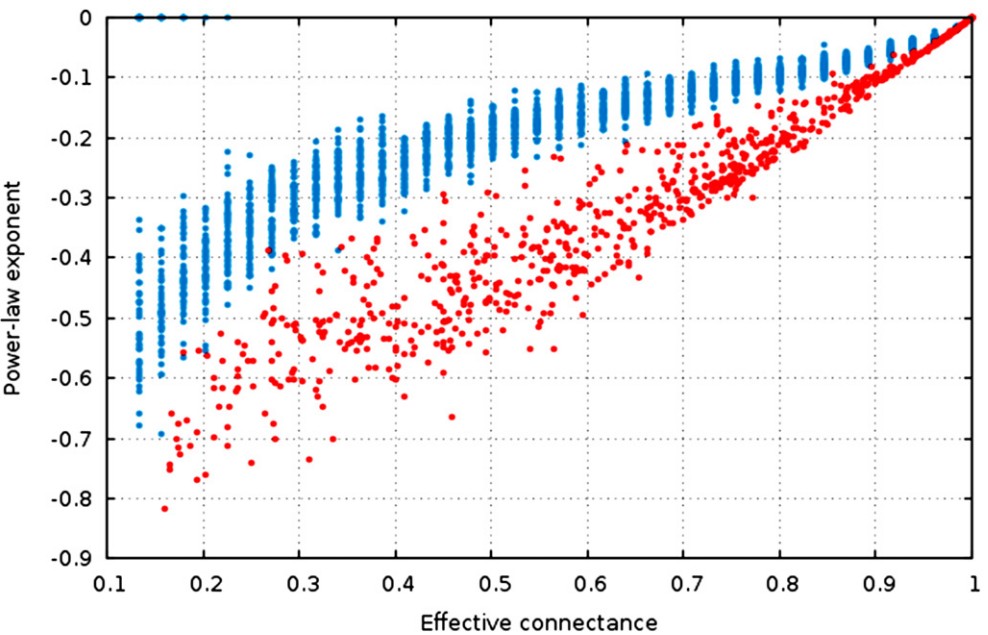

**Figure 4** The estimate of the power-law exponent increases with connectance, reaching a flat distribution for complete graphs.

deviation from the random expectation based on a drastically small sampling of the network space is real, and un-addressed; at the very least, it seems that that intensity of the replication should be dependent upon the connectance of the network one tries to replicate. On the other hand, when connectance is high, the number of unique network combinations decreases, and there is a risk to generate a number of replicates that is larger than the realized network space, thus decreasing the information content of the randomizations. To the best of our knowledge, these issues have seldom been addressed in the literature on ecological network randomization. Another problem that might be considered is that some, but not all, of these graphs will be isomorphics. For example, although there are five ways to distribute two edges between three nodes (assuming undirected edges), all five graphs can be perfectly matched to one another. This will not be the case in more complex networks, i.e., with more nodes and intermediate connectance. The consequence of this is that even though it may be possible to generate a large number of randomized networks, in a context where species identity do not matter (which is often the case in null model analyses in ecology), several of these "replicates" can actually be the same, and thus the power of null model analyses at connectances where the network space is increasingly constrained (i.e., extremely high and low connectances) should be carefully evaluated.

Second, generating null models with a low connectance is a computationally intensive task. When connectance decreases, the *realized* network space decreases faster than the *total* network space, meaning that the probability of picking a network with no 0 degree

nodes (which is simply $R_{n,l}/G_{n,l}$) goes toward zero. For this reason, classical rejection sampling (accept the random network if no nodes have no edges, reject it if not) is bound to take an unreasonable amount of time in networks with low connectance. In addition, there is a risk of selecting some particular types of networks. It makes intuitive sense that networks with extremely skewed degree distributions have less chance of being generated this way, as when a few nodes collect most of the edges, the probability than the remaining nodes each have at least one edge decreases. To the best of our knowledge, this source of bias has not received important attention in the literature. For this reason, using a purely random matrix shuffling as a starting point, then swapping interactions until no free nodes remain, seems to be a promising way to address this problem. Given the important of null-model approaches in network analysis, the generation of efficient and unbiased algorithms is a fruitful research avenue.

## CONCLUSIONS

Connectance is an extremely intuitive property of network, expressing how much of the potential interactions are realized. Through statistical reasoning and simple simulations using models of random networks, we show that for a given number of species, connectance drives (i) how many different networks exist, and (ii) some key elements of the degree distribution. We observed both among and between model quantitative changes in degree distribution along a connectance gradient. The niche model is a particularly striking example of this, with the variance in the degree distribution increasing 50-fold when connectance moves from 0.1 to 0.5. This result has practical implications for network comparisons. As descriptors of degree distribution vary with connectance, connectance should be factored out from all analyses. So as to avoid colinearity issues, this can be done by either working on the residuals of the degree distributions' property of interest. To some extent, the impact of connectance is lesser in the 0.05–0.3 range where most empirical food webs lies (although bipartite networks can have much higher connectances), but the effect is high enough that it should not be ignored: at equal number of species, networks with different connectances are expected to have different degree distributions.

Finally, this analysis raises interesting ecological questions. Early analyses focusing on degree distribution argued that ecological mechanisms were responsible for the shape of the distribution (*Vázquez, 2005*; *Fortuna et al., 2010*; *Williams, 2011*). In this contribution, we show that connectance will impose a lower and higher limit for the shape of the degree distribution. Given this information, it is time to bring the debate full-circle: is connectance the cause of observed network properties, or an emergent property of pairwise species interactions? As the latter seems far more likely, it now makes sense to focus on why some networks deviate, or not, from the expected degree distribution knowing their connectance. As the density of interaction plays such a central role in May's criteria for stability (*May, 1972*), clarifying how connectance is shaped by mechanisms regulating pairwise species interactions offers the opportunity of integrating the effects of these mechanisms up to their impact on emergent, community-wide

properties. *Okuyama & Holland (2008)* showed that in mutualistic systems, resilience (to perturbation) is affected both by degree network size and interaction strength, but also by the degree distribution and connectance of the network; we show here that degree distribution and connectance are tightly linked, and alternative approaches to the question of resilience can focus on the deviation of degree distribution knowing the connectance, rather than the "raw" degree distribution. In keeping with the results we present in the first part of the paper, *Rozdilsky & Stone (2001)* report that there is a inverted hump-shaped relationship between connectance, and the proportion of "feasible and stable" systems, with the lowest proportion of such systems being found at $Co \approx 0.5$. This strongly suggests that the same mechanisms that limit the *realized* network space may affect ecological properties, thus emphasizing the need not to discard connectance in profit of more emerging properties in future ecological network analyses.

## ACKNOWLEDGEMENTS

We thank Luis Gilarranz, Miguel Lurgi, and Enrico Rezende for comments, and Amael LeSquin for discussions on algebra.

### Funding

TP was funded by a PBEEE post-doctoral scholarship. The funder had no role in study design, data collection and analysis, decision to publish, or preparation of the manuscript.

### Grant Disclosures

The following grant information was disclosed by the authors:
PBEEE post-doctoral scholarship.

### Competing Interests

The authors declare that they have no competing interests.

### Author Contributions

- Timothée Poisot conceived and designed the experiments, performed the experiments, analyzed the data, wrote the paper.
- Dominique Gravel conceived and designed the experiments, analyzed the data, wrote the paper.

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
