# Peer review of "When is an ecological network complex? Connectance drives degree distribution and emerging network properties"

_PeerJ, doi:10.7717/peerj.251_

## Round 0.1 · original submission · Major Revisions

Your ms has been evaluated by three referees and all point out important concerns that you would need to consider before you resubmit the paper to PeerJ. Most concerns are methodological (clarify for instance, whether you used direct or undirect networks for your simulations) although you will need to check also the English style and grammar as there are a number of typographical errors. Please check also the structure of the paper and make the figure captions self-explanatory. Think also more carefully on how you want to deal with the stability issue pointed out by two of the reviewers.

Reviewer 1 ·

Basic reporting

A number of idiomatic and grammatical errors, most only irritating (e.g “a given note have a degree”), but sometimes they do compromise clarity.
Terms are used without previous reference, e.g. p.7, “the graph giant component” …? Do all networks have a giant component?
Figures are adequate to portray results, but could be improved; f.i. showing the proposed truncation value (p.10).

Experimental design

Simulations seem adequate for their purposes but limited. The paper raises the issue of stability in the Introduction and promises to interpret ecosystem properties, but does not live up to it. The most obvious indicator of stability loss- when networks break into at least two unconnected parts – could have been included easily. Instead, authors only consider the number of unconnected nodes in the resampling process.

Validity of the findings

The main finding – that connectance is unimodally correlated with various other network parameters, e.g. power-law exponent – is known, but it is convenient to have it well documented.
I was surprised by the argument that 10^3 or 10^4 permutations are insufficient (“drastically small sampling”) because the sampled space is much larger. No bias or error is demonstrated by these “small” samples (which are not samples proper anyway), so that this statement seems to ignore almost century of sampling theory.
On p.12 authors recommend a metohd (“swapping interacrtions until no free nodes remain”) which they do not seem to have used.
Bipartite networks, which have been much investigated in the ecological network literature, turn up in the conclusions without having been mentioned at all before.

Additional comments

In my judgement this paper needs substantial revision but also additional work to raise its relevance.

·

Basic reporting

In this paper, the authors show that some properties of the degree distribution of networks are driven by network connectance. They go further and say that this findings have implications for the the generation of random networks in null model analyses and the interpretation of network structure and their consequences.

Although this could be a paper applied to several areas of the sciences, the authors decided to focus on ecological networks, and for making that link they chose to use the niche model as a benchmark for testing their theory. This is ok, but I am concerned about the extent to which this is in line with the aims and scope of the journal.

In any case, and in order to make it more 'ecological' I would make reference to ecology in the title; something like: 'When is an ecological network complex?...'

The structure of the paper does not comply with the structure required by the journal. Please re-organize into the right sections. I suggest putting the methodological bits of sections 2 and 3 in a Materials and Methods sections and the results bits into a Results and Discussion section. Section 4 also in Results and Discussion, and the Conclusions. Please look at the web page for details.

References do not comply with the format (when they have three authors or less all of them should be named, within the text and in the references section). For example Dunne et al. 2002a. Please check all references and ensure they comply with this.

As a general comment, perhaps for future occasions, please enumerate the lines within each page of the manuscript. This makes easy the reviewing process.

Experimental design

This work, to my knowledge, describes original research in the area. The approach they take in terms of statistical and simulation methods is valid and sound.

However, there are some minor details I would like to see clarified:
1. why do you say (at the beginning of the third paragraph in the introduction) that researchers discard connectance in favour of degree distributions? Perhaps some references are in order.

2. in section 2 you say you are doing the statistical derivations for undirected networks and the results apply equally to directed ones. In section 3 on the other hand, you employ the niche model, which yields directed networks. Are you using directed or undirected networks for the simulations?

3. in the simulations section, you say that you experiment with 35 to M_30 interactions. At how long intervals? 1 by 1? (meaning 35, 36, 37, and so on) each 2? (35, 37, 39…), every five? (35, 40, 45…)

4. also, after describing the ER model for random graphs generation, you mention you generated 19000, and then after the niche model description you say that 500 replicates were generated for each l. What is the relationship between these numbers? Why are there different number of replicates for each type of network?

5. towards the end of page 10 you mention that 2*n is the number of links after which the power law relationship would truncate for highly connected networks. Unless networks are directed (see point 2 above), I think this value should be n. Please double check this.

Validity of the findings

The main concern I have about the validity of the findings is about the statistical significance of the results, as no statistical tests or results are presented.

For example:

the authors claim a couple of times that kurtosis is independent of connectance, while the figure (figure 3) seems to suggest some tendency, at least in niche model networks. If this claim is supported by statistical tests they should be presented, otherwise it should be clearly stated that this is your interpretation of the plot.

Another concern I have, which doesn’t have anything to do with the results but with their implications is that in section 4 the authors claim that finding networks with low connectance is very hard and will potentially take a long time to compute. Many works have used the niche model for networks generation with that number of species (30) and even more (up to a hundred) and values of connectances between 0.05 and 0.1. To my knowledge, researchers have never experienced any problems generating networks with such low connectances. I have performed experiments like this myself and have always found realistic configurations.

Additional comments

I would find a more appropriate reference for “complex” in the 6th line of the introduction.

‘Networks’ should be ‘Ecological networks’ in that same sentence.

The caption of figures 3 and 4 must specify what does each point represent. One network? An average of several? It doesn’t look like there are 19000 points on that graph, but I might be wrong.

The last word of section 4 should be changed to avenue instead of problem.

In the last paragraph of the paper you enter a very delicate discussion about the role of connectance on food web stability. I think this is out of the scope of this paper and does not fit this discussion, especially because you have not talked about stability elsewhere in the manuscript and the reader is taken by surprise when reading this. If you do not wish to remove it, you should at least rephrase it making the stability thing not the central aspect of the consequences of your current work but just another possible feature affected by it.

·

Basic reporting

No comments.

Experimental design

Everything seems fine, except for an equation which seems to be incorrect (see general comments to the authors).

Validity of the findings

No comments.

Additional comments

This study describes the major impact that connectance has on degree distribution. I find many of the results somewhat trivial, most researchers that are familiar with network analyses would immediately acknowledge that degree distribution is highly constrained by connectance.

Having said that, I agree with the authors that a formal treatment of the problem is lacking and, in this context, I find their general approach quite elegant. The problem is that they only scratch the surface of a substantially more complex problem. It is currently very difficul to quantify the realized network space for null model testing, and this approach represents a good step in that direction.

This work involves presumably with the simplest of all possible scenarios (i.e., a unipartite network with undirected 0-1 interactions), which in fact can be considerably confusing for most ecologists because, from a biolgical point of view, interactions always have a direction (which is defined a priori by the species position in rows vs columns). For example, in the first sentence of point 2 – Statistical argument, the authors claim that self-edges correspond to canibalism. However, canibalism and trophic interactions imply a direction (someone eats and someone get eaten).

This problem is not only phylosiphical, in the matrix representation of the nextwork, each new edge encompasses two new interactions because A eats B = B eats A. In other words, the connectivity that the authors are dealing with correspond to the number of interactions observed across the total size of the upper triangular matrix. Perhaps the authors might want to warn readers, particularly those not familiar with the field, about this issue (I realized that this was a problem when I tried to employ R commands "choose" and "combn" to obtain a 0-1 vector that can be readily included in a square matrix to estimate the number of different network structures). In my opinion, directed matrices that distinguish species roles in rows vs columns are mathematically more tractable (e.g., M = rows x columns interactions).

Even though one would expect results to be qualitatively under more complex scenarios, it is important to highlight and discuss other factors that are also relevant. Perhaps a major factor that is worth considering (and perhaps addressing mathematically) is the shape of the matrix. For any given network size (rows x columns), squared matrices are expected to provide the highest degrees of freedom. For highly assymetrical networks in which the number of rows and columns differ considerably, the number of combinatorials and overall degree distribution is expected to be substantially lower.

p. 4. Sentence starting with “If one node is removed from network...”: Is the equation correct? The term in the denominator (n – (n – 1)!) can be negative, which should not be the case in combinatorials. Shouldn’t it be only (n!/(n – 1)!)? Please double check.

p. 4 next line: Remove closing parenthesis in “which further simplifies to n”.

p. 4. In sentence “with at least one node of null degree”, close parenthesis.

Fig 1. I think this figure does not appropriately illustrate that "the realized network space becomes asymptotically closer to the size of the total network space when connectance increases" as stated in the caption. I am certainly aware that this is true, but perhaps the authors should consider another way of representing this. One possibility would be to include a substantially larger network or include lower connectivities in this figure.

p. 12. End of first paragraph. Absolutely. In this paragraph it is worth mentioning that this problem is not limited to degree distribution and may be pervasive in null models testing for other structural properties. For instance, I have always wondered how many matrix rearrangements exist while maintaining degree distribution constant (i.e., the fixed row, fixed column null model proposed by Gotelli, Ulrich and colleagues). My impression is that the realized space is highly constrained (Almeida-Neto et al. 2008 briefly discuss this in their NODF paper, p. 1232).

---

## Round 0.2 · accepted · Accept

Thank you for considering the most relevant concerns of the referees. I think this has notably improved the earlier version of your paper. I also believe it will be a useful contribution to network ecology.